# Position: AI Safety Must Embrace an Antifragile Perspective

**Ming Jin** [1]   **Hyunin Lee** [2]

## Abstract

This position paper contends that modern AI research must adopt an antifragile perspective on safety—one in which the system's capacity to guarantee long-term AI safety such as handling rare or out-of-distribution (OOD) events expands over time. Conventional static benchmarks and single-shot robustness tests overlook the reality that environments evolve and that models, if left unchallenged, can drift into maladaptation (e.g., reward hacking, over-optimization, or atrophy of broader capabilities). We argue that an antifragile approach—Rather than striving to rapidly reduce current uncertainties, the emphasis is on leveraging those uncertainties to better prepare for potentially greater, more unpredictable uncertainties in the future—is pivotal for the long-term reliability of open-ended ML systems. In this position paper, we first identify key limitations of static testing, including scenario diversity, reward hacking, and over-alignment. We then explore the potential of antifragile solutions to manage rare events. Crucially, we advocate for a fundamental recalibration of the methods used to measure, benchmark, and continually improve AI safety over the long term, complementing existing robustness approaches by providing ethical and practical guidelines towards fostering an antifragile AI safety community.

## 1. Introduction

**We argue that AI robustness must embrace a time-evolving perspective to mitigate the risk of black swan events and maladaptation.** Despite impressive strides in robust ML—including adversarial defenses (Goodfellow et al., 2014; Madry et al., 2017), certified robustness bounds (Wong & Kolter, 2018; Cohen et al., 2019), and safety

checks (Amodei et al., 2016; Hendrycks et al., 2021)—dominant approaches still treat robustness as a one-shot property validated on static benchmarks or static threat models before system deployment (Brendel et al., 2019; Croce et al., 2020; Miller, 2022). This snapshot perspective overlooks three fundamental realities of real-world deployment:

**1) Environments Evolve:** Mission-critical domains such as cybersecurity and critical infrastructure continually face new attack vectors, shifting user behaviors (Koh et al., 2021), and unforeseen climatic changes (Leal Filho et al., 2022). As distribution shifts become the norm rather than the exception, static robustness checks inevitably lag behind emergent threats—turning the system into a fixed target ripe for novel attacks (Quiñonero-Candela et al., 2022).[1]

**2) Incomplete World Models:** Even in unchanging conditions, black swans [2] can emerge from unknown unknowns (Lee et al., 2025). Our limited assumptions and partial information create blind spots, letting high-impact events "slip through the cracks" and catch AI systems off guard (Ibrahim et al., 2024; Dalrymple et al., 2024; Schnitzer et al., 2024). Paradoxically, over-confidence in static robustness certificates can amplify this fragility, since developers assume comprehensive safety where none truly exists (Cohen et al., 2019; Hendrycks et al., 2021).

**3) Maladaptation Over Time:** When systems are not continually challenged by new scenarios, their ability to generalize or respond to unforeseen conditions can atrophy (Sculley et al., 2015; Shafique et al., 2020; Drenkow et al., 2021; Yamagata & Santos-Rodriguez, 2024). This phenomenon—observed in natural systems—is equally relevant in data-driven AI, where over-optimization for narrow tasks leads to brittle capabilities that fail badly outside those tasks (e.g., reward hacking or over-alignment to a fixed environ-

---

[1] Recent experiences in large language models (LLMs) illustrate this urgency: jailbreak prompts often emerge within days of each guardrail update, and zero-day exploits continue to plague key infrastructure systems.

[2] We use 'black swans' to mean catastrophic events that fall outside the system's current model and are assigned near-zero probability or insufficient negative cost. While some black swans remain outright unknown unknowns, many become merely rare events once partially understood. Our goal is to systematically shrink the realm of these unknown or underweighted scenarios over time.

[1] Virginia Tech [2] UC Berkeley. Correspondence to: Ming Jin <jinming@vt.edu>.

*Proceedings of the $42^{nd}$ International Conference on Machine Learning*, Vancouver, Canada. PMLR 267, 2025. Copyright 2025 by the author(s).

ment) (Everitt et al., 2017; Lehman et al., 2020).

Collectively, these issues highlight that time-invariant robustness inadvertently promotes brittleness, lulling practitioners into a false sense of security that dissolves the moment distribution shifts or zero-day attacks appear. We therefore posit that future AI safety research must adopt an explicitly time-evolving lens—treating volatility and novelty not merely as hazards to resist but as opportunities for adaptation and growth, in line with the radical concept of antifragility (Taleb, 2010):

> "*Some things benefit from shocks; they thrive and grow when exposed to volatility, randomness, disorder, and stressors and love adventure, risk, and uncertainty. Yet, in spite of the ubiquity of the phenomenon, there is no word for the exact opposite of fragile. Let us call it antifragile. Antifragility is beyond resilience or robustness. The resilient resists shocks and stays the same; the antifragile gets better.*"

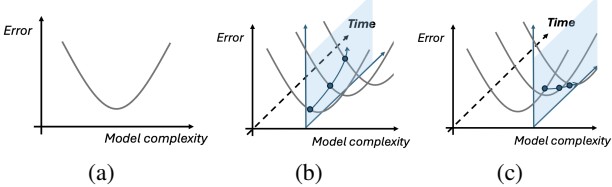

| (a) | (b) | (c) |

Figure 1: (a) Current view on robustness, (b) Our position: Fragile, (c) Our position: Antifragile. Fragile systems accumulate more vulnerabilities over time, while antifragile systems progressively reduce them as they adapt.

### 1.1. Position Statement

**Position:** *AI robustness must be reframed as a dynamic, ever-evolving property*, ensuring that each novel stressor expands the system's adaptive capacity rather than eroding it. We propose that controlled exposures to rare events, continuous monitoring for new attack scenarios, and safe stress-testing protocols become standard practice during deployment beyond training stage. Only then can we transcend the current cat-and-mouse pattern of patch-and-pray defenses and build AI systems that still remain reliable under unforeseen, significantly larger uncertainties.[3]

### 1.2. Evidence

Antifragility manifests in diverse natural and engineered systems, characterized by the ability to adapt and *improve*

---

[3]See Appendix A for how antifragility differs from established frameworks like robust MDPs, online learning, meta-learning, adversarial training, etc.

following exposure to stressors (Taleb, 2012). While the underlying mechanisms can be complex and emerge over long timescales (see Appendix F), the core principle of strengthening through challenge provides a valuable lens for AI safety (Jin, 2024). Identifying and fostering antifragile properties in AI requires moving beyond static evaluations. We contend that the community should embrace *iterative stress testing, dynamic threat modeling, and cross-team knowledge sharing* as essential practices for building systems capable of long-term adaptation and reliability.[4]

## 2. Alternative Views

**Robustness vs. Resilience vs. Antifragility.** *Robustness* typically means maintaining stable performance under known or bounded disturbances; a robust system does not break easily but also does not necessarily improve from stress. *Resilience* describes the ability to bounce back to a prior state after a shock—like a rubber band returning to its original shape. In contrast, *antifragility* involves actively thriving when confronted with volatility or novel stressors. An antifragile system leverages exposure to the unexpected to expand its safe operating regime, learning from near-failures or adversarial probes to emerge stronger rather than merely returning to baseline (See Figure 1).[5]

Skeptics might argue that frequent model updates or incremental testing already suffice (Graffieti et al., 2022; Wang et al., 2024), but such *reactive* measures still assume each upgrade re-stabilizes a system under a fixed threat landscape (Koh et al., 2021). They neglect the likelihood that unfore-

---

[4]Glimpses of behavior related to antifragility can be seen in AI/ML, though often representing partial aspects. For instance, certain meta-learning (Vilalta & Drissi, 2002; Finn et al., 2019; Vettoruzzo et al., 2024) and few-shot adaptation (Sung et al., 2018; Wang et al., 2020) algorithms exhibit rapid adjustment to new constraints after training across diverse tasks. Theoretical and empirical results, such as those by Khattar et al. (2023), show that broader scenario exposure can accelerate safe adaptation in new environments. Also, practical processes in the AI safety ecosystem demonstrate iterative refinement driven by dynamic challenges: benchmarks like Adversarial NLI (ANLI) (Nie et al., 2020) improve models via rounds of adversarial data collection; platforms like Dynabench (Kiela et al., 2021) use human-generated adversarial examples for continuous model assessment and improvement; and industry red-teaming efforts embody cycles of stress-testing and refinement (Ganguli et al., 2022). Furthermore, research is actively exploring internal mechanisms, such as structured backtracking for error recovery in reasoning and safe generation (Sel et al., 2025c;b; Zhang et al., 2025), indicating pathways towards building resilience directly into AI architectures. While valuable, these examples often represent specific mechanisms (like rapid adaptation or iterative patching) rather than the full scope of antifragility, which encompasses proactive strengthening and expanding operational boundaries in response to unexpected stressors.

[5]While we refer to 'black swan' events per Taleb's usage (rare, severe, unanticipated), in practice, antifragile methods also address smaller or more routine shifts that arise in deployment.

seen vulnerabilities and distribution shifts can emerge faster than any patch cycle (Amodei et al., 2016).

Others may contend that a static risk appetite is an acceptable trade-off for simpler certifications (Varshney, 2016; Marcus, 2018). However, this overlooks once-in-a-decade black swan events whose catastrophic impact is almost guaranteed over long horizons (Taleb, 2010).

**Antifragility Complements Robustness.**  Antifragility itself can face skepticism, since reliability and security generally require caution, not experimentation (García & Fernández, 2015; Brundage et al., 2018; Hemphill, 2020). We do not propose eliminating traditional safeguards or embracing reckless disorder; rather, we advocate selective harnessing of volatility within safety constraints (García & Fernández, 2015). Purposeful stress-testing, conducted in simulated or small-scale environments (Peng et al., 2018; Tobin et al., 2017), allows recoverable failures that ultimately strengthen system capabilities (see Appendix G). In critical domains like healthcare or cyber-physical infrastructures, such controlled measures can reveal blind spots without endangering real-world operations (Parisi et al., 2019).

Crucially, antifragility does not oppose but *complements* robustness and resilience. Robustness, whether through redundancy or certified defenses, provides an essential safety net against known or bounded disturbances, enabling the guided exploration that antifragility requires (Madry et al., 2017; Cohen et al., 2019). Resilience ensures systems can recover from transient shocks/near-failures. Antifragility builds upon these foundations, focusing on how systems can learn, adapt, and *emerge stronger* when confronted with novel stressors or surprises, especially those that push beyond the boundaries of existing robustness guarantees (Taleb, 2010; Mullainathan & Spiess, 2017).

The appropriate balance between these paradigms is context-dependent: traditional robustness may suffice in highly stable, predictable environments, whereas antifragility becomes increasingly vital in open-ended, dynamically evolving domains (e.g., LLMs, cybersecurity) where unforeseen challenges are the norm (Koh et al., 2021). Antifragility extends beyond immediate reactions; it operates on longer timescales, treating volatility not just as a threat but as information to drive adaptation. By systematically learning from smaller, manageable (near-)failures, it offers a path towards systems that are *unusually* robust precisely because they can preempt or better handle the inevitable rare, large-impact black swan events (Amodei et al., 2016; Hendrycks et al., 2021). In essence, robustness sustains today's operations; antifragility invests in tomorrow's adaptability and upside. Appendix A further discusses connections to related paradigms like lifelong learning and meta-learning.

## 3. The Inevitability of Black Swan Events

This section argues that catastrophic failures, hereafter called *black swan events*, are *inevitable* in complex AI systems. Note that the general principle discussed here (shocks that reveal new transitions or reward structures) remains valid in more complex domains (see Appendix D).

Consider a Markov Decision Process (MDP) $\mathcal{M} = \langle \mathcal{S}, \mathcal{A}, P, R, \rho, \gamma, T \rangle$ with a finite (or countable) state space $\mathcal{S}$, a finite (or countable) action space $\mathcal{A}$, a transition function $P : \mathcal{S} \times \mathcal{A} \to \Delta(\mathcal{S})$, mapping each state-action pair to a distribution over next states, a reward function $R : \mathcal{S} \times \mathcal{A} \to \mathbb{R}$, an initial state distribution $\rho \in \Delta(\mathcal{S})$, a discount factor $\gamma \in [0, 1]$, and a horizon $T \in \mathbb{N}$ (finite or infinite).[6] The value function of a policy $\pi$ under $\mathcal{M}$ is

$$V_{\mathcal{M}}(\pi) = \mathbb{E}\Big[\sum_{t=0}^{T} \gamma^t R(s_t, a_t) \,\Big|\, s_0 \sim \rho, \pi, P \Big].$$

We distinguish:

- The *real-world MDP*, $\mathcal{M} = \langle \mathcal{S}, \mathcal{A}, P, R, \rho, \gamma, T \rangle$, representing the *true* environment in which AI systems are deployed.

- The *agent's (or community's) perceived MDP*, $\mathcal{M}^{\dagger} = \langle \mathcal{S}, \mathcal{A}, P^{\dagger}, R^{\dagger}, \rho, \gamma, T \rangle$, distorted by imperfect or biased understanding of transitions and rewards. Here, $P^{\dagger}$ and $R^{\dagger}$ reflect the *subjective* or *dominant* beliefs of some research group, company, or broader community.

**Distortion and Misalignment.**  Following (Lee et al., 2025), we let

$$P^{\dagger}(\cdot \mid s, a) = w\big(P(\cdot \mid s, a)\big), \quad R^{\dagger}(s, a) = u\big(R(s, a)\big),$$

where $w(\cdot)$ and $u(\cdot)$ are *probability* and *value* distortion functions, respectively (see Appendix B for function definitions). These reflect, for example, the tendency for typically perceiving losses as more significant than equivalent gains and often underestimating the likelihood of rare events (Kahneman & Tversky, 2013; Fennema & Wakker, 1997).

**Robustness Gap.**  We say the agent holds a *perceived MDP $\mathcal{M}^{\dagger}(t)$* at time $t$, which may be updated over time as the agent gathers new evidence. Let $\Delta(t)$ measure how the *best* policy in the perceived MDP can be quite bad compared to the *true-optimal* policy in the *true* environment $\mathcal{M}$:

$$\Delta(t) = V_{\mathcal{M}}(\pi^{\star}) - V_{\mathcal{M}}(\pi_t^{\dagger}),$$

---

[6] We use MDPs primarily as a conceptual tool to model the sequential, interactive nature of AI systems and the potential gap between a system's perceived model and the complexities of the real environment, rather than as a literal implementation requirement for all systems.

where $\pi_t^\dagger = \arg\max_\pi V_{\mathcal{M}^\dagger(t)}(\pi)$ and $\pi^\star = \arg\max_\pi V_{\mathcal{M}}(\pi)$.

Then, we define *a black swan event* as the agent encountering certain state $s \in \mathcal{S}$ during planning.

**Definition 3.1** (Black Swan Event (Catastrophic Failure)). A *black swan event* is realized if the *robustness gap* $\Delta(t)$ is large for some $t$, i.e., there exists a policy $\pi_t^\dagger$ deemed (near-)optimal in $\mathcal{M}^\dagger(t)$ whose real-world value differs greatly:

$$\Delta(t) \gg 0.$$

When such mismatches occur, the agent may execute $\pi^\dagger$ in good faith, yet unexpectedly encounters catastrophic states. In this case, the trajectory under $\pi^\dagger$ visits states $s$ with $R(s,a) \ll 0$, yet in the agent's model, $R^\dagger(s,a) = u(R(s,a))$ is (mis)perceived as far less severe. This is a black swan in the sense of (Taleb, 2010), but in an MDP formalism, it is a *robustness gap* with high-severity states unaccounted for.

### 3.1. Emergence of Black Swan Events

We adapt three theorems from (Lee et al., 2025), which show that in complex multi-step settings, a non-zero gap is unavoidable.

**Theorem 3.2** (Trivial Cases Without Black Swan). *If $|\mathcal{S}| = 2$ or $T = 1$, then no black swan event occurs, i.e., $\Delta(t) = 0$ for all $t$.*

Intuitively, the result is due to that the agent's perceived model $\mathcal{M}^\dagger(t)$ cannot yield a large gap because the environment is too simple or single-step.

**Theorem 3.3** (Multi-State, Multi-Step Gaps). *In any environment with $|\mathcal{S}| \geq 3$ and $T \geq 2$, one can construct $P^\dagger$ and $R^\dagger$ such that an optimal policy in $\mathcal{M}^\dagger$ is* catastrophically *suboptimal in $\mathcal{M}$. A black swan event (large gap) thereby occurs with non-zero probability.*

Together, Theorems 3.2 and 3.3 show that in realistic, multi-step AI deployments, some fraction of the perceived $P^\dagger$, $R^\dagger$ will *miss or downplay* rare-but-possible transitions. As a result, *black swan events are inevitable* whenever the environment is rich enough to support severely negative, low-probability outcomes.

Below is a corollary that links back to the *robustness gap* and shows that why $\Delta(t)$ cannot vanish. We introduce a measure of environmental sensitivity

$$\Delta_1(\pi) := \big| V_{\mathcal{M}}(\pi) - V_{\mathcal{M}^\dagger}(\pi) \big|,$$

i.e., the absolute difference in value *of the same policy* $\pi$, when evaluated in the real MDP vs. the perceived MDP. We list the following conditions:

**(1)** *Small environmental sensitivity for $\pi^\star$.*
There is a nonnegative constant $\epsilon^\star \geq 0$ such that

$$\Delta_1(\pi^\star) = \big| V_{\mathcal{M}}(\pi^\star) - V_{\mathcal{M}^\dagger}(\pi^\star) \big| \leq \epsilon^\star.$$

(In words, the real-optimal policy $\pi^\star$ is *not* severely misperceived.)

**(2)** *Large environmental sensitivity for $\pi^\dagger$.*
There is a strictly positive constant $c^\dagger > 0$ such that

$$\Delta_1(\pi^\dagger) = \big| V_{\mathcal{M}}(\pi^\dagger) - V_{\mathcal{M}^\dagger}(\pi^\dagger) \big| \geq c^\dagger.$$

(This captures the idea that the agent's chosen policy $\pi^\dagger$ looks good in $\mathcal{M}^\dagger$ but is severely misrepresented compared to the real world.)

**(3)** *Margin $\delta^\dagger$ in the perceived MDP.*
In the distorted MDP $\mathcal{M}^\dagger$, the chosen policy $\pi^\dagger$ does not differ significantly from $\pi^\star$:

$$V_{\mathcal{M}^\dagger}(\pi^\dagger) - V_{\mathcal{M}^\dagger}(\pi^\star) \leq \delta^\dagger.$$

(This ensures $\pi^\dagger$ is not significantly better than $\pi^\star$ when viewed through the distorted lens.)

Conditions (1) and (3) can be viewed as the robustness property of the truly optimal policy $\pi^*$, whose performance is similar in both the distorted and the true MDP and also does not differ significantly from $\pi^\dagger$ in the distorted MDP. Condition (2) is justified by Theorem 3.3.[7]

**Corollary 3.4** (Robustness Gap Lower Bound). *Suppose Conditions (1)–(3) above hold at time $t$. If*

$$c^\dagger > \delta^\dagger + \epsilon^\star,$$

*then the robustness gap is positive:*

$$\Delta(t) \geq c^\dagger - \big( \delta^\dagger + \epsilon^\star \big) > 0$$

cannot *vanish.*

---

[7] Furthermore, by Theorem 5.1 of (Lee et al., 2025), the mismatch $\Delta_1(\pi^\dagger)$ admits a lower bound of the form

$$\Delta_1(\pi^\dagger) \geq \Omega\Big( \epsilon_{bs}^{\min} \times C_{bs} \Big),$$

where $\epsilon_{bs}^{\min} > 0$ is the minimal (nonzero) probability of some *rare* but catastrophic states in the real MDP $\mathcal{M}$; $C_{bs} > 0$ measures how severely the agent *distorts* negative rewards or probabilities (e.g. ignoring or underestimating black swan states). Hence taking

$$c^\dagger := \Omega\big( \epsilon_{bs}^{\min} C_{bs} \big) > 0$$

makes explicit that black-swan events guarantee a nonzero gap between $V_{\mathcal{M}}(\pi^\dagger)$ and $V_{\mathcal{M}^\dagger}(\pi^\dagger)$.

The necessary condition of Corollary 3.4, namely $c^\dagger > \delta^\dagger + \epsilon^*$, indicates that $\pi^\dagger$ exhibits greater environmental sensitivity than $\pi^*$ (Conditions (1) and (2)), under the additional assumption that $\pi^\dagger$ is close to $\pi^*$ (Condition (3)). This observation highlights a strategy for reducing the robustness gap. Since the gap between $\pi^\dagger$ and $\pi^*$ is already small, one can further lessen $\pi^\dagger$'s perception sensitivity by narrowing the difference between $\mathcal{M}^\dagger$ and $\mathcal{M}$. Concretely, re-learning (or re-weighting) the functions $w$ and $u$ so that $\mathcal{M}^\dagger$ converges to $\mathcal{M}$ can effectively reduce the overall robustness gap. If the agent *never* reweights those black swan transitions/rewards—that is, if $\mathcal{M}^\dagger(t)$ maintains $w\big(P(\cdot \mid s,a)\big) \approx 0$ or $R(s,a) << u\big(R(s,a)\big) < 0$ for some truly severe negative states—the agent's *best policy* in the perceived MDP $\mathcal{M}^\dagger(t)$ inevitably incurs a large mismatch vs. $\mathcal{M}$. No amount of fine-tuning the model on *already-known data* collected within the agent (or the community)'s model changes this if the black swan states remain systematically discounted as "impossible." Therefore, $\Delta(t)$ remains underbounded away from zero, reflecting unavoidable catastrophic failures when $\pi^\dagger(t)$ eventually visits those black-swan states.

## 3.2. Why Black Swan Events Are Inevitable

We highlight two core reasons such large-gap, catastrophic failures will *always* arise, as detailed in Subsection 3.1, in sufficiently complex AI systems, regardless of how intensively we test them under a *static* or *consensus* view.

**Fundamental Distortions in Human and Community Reward Perception** A key insight is that, even among AI researchers themselves, there is no universal agreement on whether certain "extreme risk" scenarios are realistic (Bostrom, 2018; Marcus, 2018; Ord, 2020). For instance, some groups believe advanced AI must be halted or heavily restricted to avoid doomsday scenarios (Carlsmith, 2022); others see AI as merely a tool, with negligible existential threat (Silver et al., 2021; Brynjolfsson & Mitchell, 2017). This points to the society's optimism/pessimism split. One faction overestimates short-term gains, setting $u(\cdot)$ to emphasize innovation reward while discounting rare catastrophic costs, (Team, 2021; Krakovna et al., 2020) whereas a safety-oriented faction sets $u(\cdot)$ to heavily penalize such risks (Soares et al., 2015). At least one faction's distortion must be "incorrect," indicating $R^\dagger$ systematically departs from $R$. Given such persistent disagreement, no consensus "true reward" emerges (Knox et al., 2023; Booth et al., 2023); *some* part of the field underestimates or misjudges negative outcomes, implying a permanent $\Delta(t) > 0$ scenario.

**Blind Spots in Transition Probabilities** Even when rewards align, new attack modes or hidden environment transitions repeatedly emerge (Goodfellow et al., 2014; Huang

et al., 2011; Papernot et al., 2018). For instance, for adversarial ML, the *small-perturbation attacks* (Szegedy et al., 2014) was a revelation moment for the entire field, followed by a series of novel attack modes such as *physical adversarial attacks* (Kurakin et al., 2018), backdoor/trojan attacks (Gu et al., 2017; Liu et al., 2018), etc., each of which were deemed "unlikely" or "impractical" until papers demonstrated easy triggers. Each discovery reveals a gap: the real transition $P(\cdot \mid s,a)$ allowed an unexpected path, while $w\big(P(\cdot \mid s,a)\big) \approx 0$ in the community's model (Kurakin et al., 2018; Tramer et al., 2020). *Hence large gaps keep arising* as new states or transitions come to light.

Thus, the environment's support of black swan transitions (and the agent's refusal or inability to assign them proper probability/reward weighting) leads to an *inevitable* risk of catastrophic failures, consistent with the broader notion of "black swans" in open-ended AI systems (Taleb, 2010; Wei et al., 2022).

*Comments.* One might question whether restricting to an MDP formalism is too simplistic, especially in partially observed domains or multi-agent interactions (Kaelbling et al., 1998; Buşoniu et al., 2008). Our point is that even in a simple MDP, black swan events are inevitable once we allow multi-step dynamics and rare transitions. This implies that in more complex settings—with partial observability (Spaan, 2012), uncertain reward structures, or high-dimensional sensor data (Mnih et al., 2015) —black swans are, if anything, more likely. The takeaway is: If black swan events can arise in a simple MDP, they certainly remain a concern in any richer real-world environment (Sutton & Barto, 2018; Leike et al., 2017).

## 3.3. Real-World AI Safety Suffers from Black Swans

**Large Language Models (LLMs).** Despite extensive red-teaming and iterative patching, modern LLMs (e.g. ChatGPT, Bard) continue to exhibit jailbreak vulnerabilities (Zou et al., 2023; Tedeschi et al., 2024; Wei et al., 2024):

- *Robustness for a While*: Early tests may suggest the model is safe against certain adversarial prompts (Solaiman et al., 2019; Brown et al., 2020).

- *Sudden Loopholes*: In the wild, users discover new, unanticipated *attack* or *prompt* configurations that circumvent guardrails (Zou et al., 2023).

This perfectly illustrates a black swan scenario: a small-probability exploit that the gap between the development community and the real environment (Ganguli et al., 2022). Compounding this challenge, many AI safety benchmarks are highly correlated with general capabilities rather than measuring distinct safety properties, potentially enabling a safetywashing phenomenon where capability improve-

ments are misrepresented as safety advancements (Ren et al., 2024).

**Critical Infrastructure (Physical/Cyber).**  Pentesting has long been standard in critical infrastructures (power grids, water systems, etc.) (Cárdenas et al., 2008), including cross-domain (physical + digital) attacks (Loukas, 2015). Yet, as more components become interconnected (IoT devices, remote sensors), unforeseen vulnerabilities arise that security teams did not anticipate. Real-world cyberattacks continue to evolve faster than one-off testing can accommodate, reflecting repeated large gaps between the tested model vs. the actual risk landscape (Gupta & Shukla, 2016).

Thus, new high-severity intrusions keep emerging, underscoring that Black Swan failures inevitably appear in sufficiently large or complex systems (Zarpelão et al., 2017).

## 4. Fragility, Anti-Fragility, and a Regret-Based View

One might ask: *How can we formally tell whether a system is fragile in the face of these black swan failures?*

To begin with, it is not clear how, as fragility lies on a scale, i.e., the system may be robust to a certain point then breaks (Taleb, 2012; Nguyen et al., 2015). However, when adding the time dimension, a binary classification is sensible (see Figure 1 (b) and (c)). A purely static claim—"the model passed certification"—may reflect a fragility mindset. Over a longer horizon, a system either reduces its *robustness gap* $\Delta(t)$ or remains vulnerable to new incidents (Amodei et al., 2016; Hendrycks et al., 2021; Taleb, 2010).

### 4.1. Regret-Based Definition

We formalize a two-timescale process, where the *fast loop* (policy iteration $\pi_{t,i}$ to operate on the current community model) and *slow loop* (the community's model updates over time):

- **t** (the **slow loop** index): tracks how the community's perceived MDP, $\mathcal{M}^\dagger(t)$, evolves at discrete "community-update" times $t = 1, \ldots, T$.

- **i** (the **fast loop** index): tracks the internal optimization or stress-test iterations $\pi_{t,i}$ for $(i = 1, \ldots, N)$ performed within $\mathcal{M}^\dagger(t)$ before the next slow-loop update occurs.

**Within-Model Regret (Fast Loop).**  At each slow-loop step $t$, the community solves a sequence of internal policy iterations $\pi_{t,1}, \pi_{t,2}, \ldots, \pi_{t,N}$ in the *perceived* MDP $\mathcal{M}^\dagger(t)$. To measure suboptimality *within* this fixed model, we compare against the best policy in a chosen comparator MDP,

denoted $\widetilde{\mathcal{M}}$:

$$\Delta(t, i, \widetilde{\mathcal{M}}) = V_{\widetilde{\mathcal{M}}}(\tilde{\pi}) - V_{\widetilde{\mathcal{M}}}(\pi_{t,i}),$$

where $\tilde{\pi} = \arg\max_\pi V_{\widetilde{\mathcal{M}}}(\pi)$. This captures how far $\pi_{t,i}$ lags behind the $\widetilde{\mathcal{M}}$-optimal policy $\tilde{\pi}$. One may choose $\widetilde{\mathcal{M}}$ to be the *true environment* $\mathcal{M}$ (for genuine robustness-gap comparisons), or the *best feasible model* the community can construct (a "best-effort" comparator) to reflect the practical limitations due to technology, social factors, etc.

We define the *average within-model regret* across $i = 1, \ldots, N$ iterations at slow-loop step $t$:

$$\Delta(t, \widetilde{\mathcal{M}}) = \frac{1}{N} \sum_{i=1}^{N} \Delta(t, i, \widetilde{\mathcal{M}}).$$

A standard *fast-loop convergence requirement* is that $\Delta(t, \widetilde{\mathcal{M}}) = o(N)$, meaning the community eventually finds a near-optimal policy *within* its current perception.

**Community-Model Regret (Slow Loop).**  Every time the community observes new attacks, new states, or other evidence that invalidates $\mathcal{M}^\dagger(t)$, it may update to $\mathcal{M}^\dagger(t + 1)$. Let $\widetilde{\mathcal{M}}_t$ be the *comparator MDP* at time $t$; it can either remain static (e.g. $\widetilde{\mathcal{M}}_t = \mathcal{M}$) or shift if the real environment itself changes. We define a *dynamic* slow-loop regret:

$$\mathcal{R}(T; \{\widetilde{\mathcal{M}}_t\}_{t=1}^T) = \frac{1}{T} \sum_{t=1}^{T} \Delta(t, \widetilde{\mathcal{M}}_t), \qquad (1)$$

where $\Delta(t, \widetilde{\mathcal{M}}_t)$ is the average fast-loop gap at time $t$ (see above). Thus, $\mathcal{R}(T; \{\widetilde{\mathcal{M}}_t\})$ measures how quickly the community's modeling process handles new or changing states and attacks.

The regret (1) can be either static regret if we choose $\widetilde{\mathcal{M}}_t = \mathcal{M}$ for all $t$, or dynamic regret if $\widetilde{\mathcal{M}}_t$ itself evolves over time (e.g. a sequence of best-effort models). In this case, one often introduces a measure of volatility, $\sum_{t=1}^{T-1} \mathrm{d}(\widetilde{\mathcal{M}}_t, \widetilde{\mathcal{M}}_{t+1}) \leq \mathcal{V}(T)$ for some distance function $\mathrm{d}$, where $\mathcal{V}(T)$ bounds how much $\widetilde{\mathcal{M}}_t$ can drift. In high-stakes AI safety, where adversaries or new states can appear abruptly, $\mathcal{V}(T)$ can be large to push the changes of best-effort community models.

We now formally define *anti-fragility* and *fragility* in terms of the dynamic regret $\mathcal{R}(T; \{\widetilde{\mathcal{M}}_t\})$ introduced in (1).

**Definition 4.1** (Fragility and Anti-Fragility)**.**  Consider a sequence of model updates $\mathcal{M}^\dagger(1), \ldots, \mathcal{M}^\dagger(T)$ and corresponding comparator MDPs $\{\widetilde{\mathcal{M}}_t\}_{t=1}^T$. Let $\mathcal{R}(T; \{\widetilde{\mathcal{M}}_t\})$ be the dynamic regret defined in Equation (1).

1. **Anti-Fragile:** We say the system is *anti-fragile* if $\mathcal{R}(T; \{\widetilde{\mathcal{M}}_t\})$ decreases in $T$ (up to statistical or random fluctuations), implying the community actively

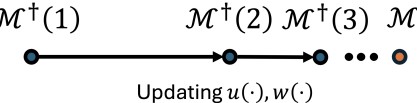

Figure 2: Update distortion function $u, w$ to be identity function is a way to attain anti-fragility.

refines its model and lowers the overall robustness gap over time, even if the system began in a flawed or incomplete state (refer to Figure 2 for an illustration of one method to achieve antifragility).

2. **Fragile:** We say the system is *fragile* if $\mathcal{R}\big(T; \{\widetilde{\mathcal{M}}_t\}\big)$ increases in $T$, meaning the system's unaddressed vulnerabilities accumulate, leading to larger regret (or gap) over time.[8]

**On the "Robust" Middle-Ground.** In reality, since threats, reward hacks, and maladaptation accumulate over time, we seldom observe a true "middle ground" of a consistently *robust* system, where $\mathcal{R}\big(T; \{\widetilde{\mathcal{M}}_t\}\big)$ remains approximately the same. Rather, it seems systems tend to drift in one of two directions:

- **Anti-Fragile Route:** The community *meticulously* reduces the robustness gap with iterative, proactive refinements (slow-loop model updates, targeted stress tests, etc.). This process continuously folds new vulnerabilities into the agent's perception model, thereby driving dynamic regret down.

- **Fragility Trap:** The system remains static or reactive, caught in a cat-and-mouse cycle of black swan events— each discovery prompts ad-hoc fixes, but no overall closure of the gap occurs. Over time, small vulnerabilities accumulate into large, disruptive failures.

**Measuring Anti-Fragility** Although anti-fragility (improving under stress) is intuitive, it can be tricky to measure in practice. In our *dynamic regret* framework, one checks whether the overall robustness gap decreases over time. In concrete systems, this may leverage: 1) High-fidelity simulations or digital twins to repeatedly test extreme scenarios; 2) Partial or counterfactual feedback to handle rare but high-impact failures (e.g. catastrophic incidents, malicious exploits); and 3) Statistical ML methods to estimate uncertainty around the gap, based on limited stress-test data.

**Comparison to Existing Definitions.** Early formalizations of (anti-)fragility in Taleb & Douady (2013); Taleb &

West (2023) focus on how a random variable's payoff distribution changes under volatility—essentially a *distribution-centric* view on the tail behavior. While this captures single-step or static risk sensitivity, it does not directly account for the iterative, community-based AI safety context where models evolve, attacks emerge, and policies adapt.

By contrast, the dynamic regret-based definition above follows the learning-theoretic perspective of Jin (2024), who propose using an online or nonstationary decision-making framework to capture changing environments and continuous model refinement. Our approach interprets their loss function specifically as the *robustness gap* arising from the community's perception model versus newly revealed real-world states or threats.

We also point out that Jin (2024) revealed a key assumption in the online learning literature that poses theoretically unachievable sublinear regret—and hence full antifragility— under adversarial or highly nonstationary conditions. Specifically, many lower bounds exploit unpredictable shifts (e.g., (Besbes et al., 2015; Zhang et al., 2018)) or information-theoretic limits on unstructured function classes (Campolongo & Orabona, 2021; Baby & Wang, 2021). As Jin (2024) noted, letting the agent *adapt on the fly* partially mitigates such issues, but cannot guarantee a dynamic regret of zero. From a black swan perspective, we agree that *perfect* avoidance is unrealistic; the practical goal is to *decrease* the gap over time, reflecting a core principle of antifragility— the process of improving rather than eradicating rare failures altogether.

Hence, designing *anti-fragile* methods to systematically shrink the black swan gap remains an *open problem* in high-stakes AI safety, where feedback can be sporadic (catastrophic mistakes are rare but high-impact), the environment may shift abruptly, and adversaries can adapt faster than the agent's slow loop can track.

## 5. Ethical and Practical Guidelines for an Anti-Fragile AI Safety Community

This section integrates two perspectives: (1) *ethical considerations* for a community-wide anti-fragile mindset (Jobin et al., 2019; Floridi, 2021; Hagendorff, 2020) and (2) a *practical checklist* outlining warning signs of fragility and concrete steps to embrace anti-fragility (Ayling & Chapman, 2022; Amodei et al., 2016; Hendrycks et al., 2021).

### 5.1. Ethical Considerations

Adopting an *anti-fragile* approach in AI safety means focusing on iterative stress-testing, open vulnerability disclosure, and collaborative refinement of our collective perception model $\mathcal{M}^{\dagger}(t)$ (Brundage et al., 2018; Papernot et al., 2018). This shift raises unique ethical issues:

---

[8]Refer to Figure 1 (b) and (c) for illustrations of fragile and antifragile classifications.

- *Selective Disclosure vs. Collective Safety.* Publicizing new exploits or black swan risks accelerates community fixes but could guide malicious actors (Familoni, 2024). Responsible disclosure policies should ensure timely mitigation while preventing premature leaks.

- Large-scale cross-team testing (with ethicists, social scientists, domain experts) helps avoid blind spots (Whittlestone et al., 2019), ensuring we do not overlook certain user communities or demographic groups (Mitchell et al., 2019).

- *Data Sensitivity & Privacy.* Logs of failures or near misses expedite learning, but often contain private or sensitive details. Ethical data governance is crucial to protect participants while enabling community-wide improvements (Mittelstadt, 2019).

- Intentional large-scale adversarial trials risk disrupting deployed systems or users; sandboxing and simulation minimize real-world damage and liability (Leike et al., 2017; Carlsmith, 2022). If a test can only be done in live settings, ensure user consent and fallback mechanisms (Wei et al., 2022).

- *Resource Inequities.* Anti-fragile research frequently demands high compute and specialized skills, which not all labs can afford (Raji et al., 2020). Open testbeds and collaborative funding can level the playing field, preventing an elite few from dominating black-swan discovery (Ayling & Chapman, 2022).

Ultimately, AI safety aims to curb existential or multi-decade threats (Ord, 2020). A short-term or reactive stance might neglect truly catastrophic scenarios. Community-wide iterative models must keep the bigger picture in focus, avoiding an arms-race mentality and championing global safety frameworks (Bostrom, 2018; Russell, 2022).

### 5.2. Practical Checklists

We provide a short checklist to help researchers and practitioners spot *fragility* in AI robustness claims, followed by recommended actions to move toward *anti-fragile* approaches (Taleb, 2012; Soares et al., 2015). Our goal is not to be exhaustive, but to offer concrete, easily identifiable red flags and positive steps.

**Red Flags of Fragility**

1. **"We passed *the* robustness test!"** A single static test suite or benchmark often cannot capture open-ended adversaries or shifting environments (Goodfellow et al., 2014; Madry et al., 2017). If a method rests on *one* final certification without ongoing re-evaluation, fragility

is likely. Similarly, claiming completeness almost always risks overconfidence due to rare and novel modes of failure emerging post-deployment (Papernot et al., 2018).

2. **No mention of *temporal* or *iterative* updates.** If the approach disregards how real-world threats (or the model itself) evolve over time, it is prone to future blind spots (Koh et al., 2021).

3. **Neglecting adaptive or adversarial feedback loops and lack of open-ended stress testing.** Threat actors usually adapt. A purely one-step analysis—treating adversaries as fixed—often leads to major gaps (Tramer et al., 2020). Similarly, if a team does not invite outside testers or does not encourage adversarial probes beyond known test cases, it is failing to expand its perception model.

4. **No post-deployment feedback.** Systems that do not incorporate monitoring or slow-loop model updates of $\mathcal{M}^{\dagger}(t)$ over time risk abrupt catastrophes (Amodei et al., 2016; Hendrycks et al., 2021).

*In short, any one-time or closed-world claim of "robustness" can signal a fragile mindset.*

**Strategies Toward Anti-Fragility**   Conversely, here are steps and strategies that promote *anti-fragile* practices:

1. *Foster Internal Resilience Mechanisms.* Design AI systems not just to avoid errors, but to handle internal failures gracefully. Actively explore and integrate mechanisms such as structured backtracking for error recovery in reasoning, or safe state reversion during generation (e.g., (Sel et al., 2024; 2025b; Zhang et al., 2025)), allowing systems to manage and potentially learn from operational mistakes rather than consistently succumbing to catastrophic failures.

2. *Slow-Loop Updates of Community Model.* Rather than finalize $\mathcal{M}^{\dagger}$, anticipate new states, vulnerabilities, and adversarial behaviors. Integrate each discovery into the model pipeline (Papernot et al., 2018; Lee et al., 2024b).

3. *Multi-Phase Collaborative Testing.* Incorporate adversarial prompts (for LLMs) or cross-domain intrusion (for infrastructure) on a regular basis (Tramer et al., 2020; Cárdenas et al., 2008). Publicize test protocols, invite external red teams. Cross-organization collaboration often reveals blind spots faster (Brundage et al., 2018).

4. *Quantify Time-Evolving Performance.* Move beyond a single pass/fail metric. Track *dynamic regret* across

repeated policy and model updates, measuring how the gap shrinks or grows over time (Taleb, 2012).

5. *Accommodate Partial & Sporadic Feedback.* Lean on robust anomaly detection, fallback modes, or simulation expansions to preempt black swans in data-sparse scenarios (Leike et al., 2017).

6. *Embrace "Impossible" States and Invest in Safe Exploration.* Use sandboxed exploration to push beyond typical distributions, bridging knowledge gaps safely (Mullainathan & Spiess, 2017; Parisi et al., 2019).

7. *Periodic Policy Reviews.* Production systems should *never* be fire-and-forget. Schedule re-verifications, re-run adversarial checks, and refine environment assumptions at set intervals (Sutton & Barto, 2018).

8. *Adaptive Benchmarks.* Replace static leaderboards with evolving challenge suites that incorporate new exploits or environment shifts (Koh et al., 2021). Explicitly model multi-step or adaptive attackers by shifting from "the attacker is static" to "the attacker gains new capabilities over time" in threat model, forcing the slow loop to adapt (Tramer et al., 2020; Goodfellow et al., 2014; Lee et al., 2024a).

9. *Bridging Academia, Industry, and Policy.* Collaboration among AI labs, regulators, and stakeholders to ensure ongoing disclosure of vulnerabilities. Possibly create an AI version of CVE (Common Vulnerabilities and Exposures) so the knowledge of black swans accumulates and is re-checked systematically (Ayling & Chapman, 2022; Familoni, 2024).

**Concluding Note.** While our discussion remains high-level, we recognize that practitioners may require tailored protocols for specific domains (Leike et al., 2017; Khetarpal et al., 2022). We encourage future work to explore specialized best practices, software frameworks, and standardized iterative protocols for stress testing and environment expansion (Hendrycks et al., 2021; Amodei et al., 2016). For now, our chief aim is to recalibrate the AI safety conversation toward time-evolving adaptation, continuous stress-testing, and the principle that systems can learn and improve from rare, high-impact shocks—rather than viewing them solely as adversities to be contained. Addressing the significant practical challenges of implementing these ideas robustly and safely, especially in resource-constrained settings or high-stakes applications, remains a crucial direction for future research.

## Acknowledgements

The authors are grateful for the support by the NSF Safe Learning-Enabled Systems Program under grant NSF #2331775. M. Jin also acknowledges the general support by Deloitte AI Center of Excellence, the Amazon-Virginia Tech Initiative for Efficient and Robust Machine Learning, and the Commonwealth Cyber Initiative for this work.

## Impact Statement

This position paper advocates for an antifragile approach to AI safety, aiming to enhance long-term system reliability by enabling AI to learn and strengthen from encounters with novel stressors and rare events. While this paradigm promises more resilient AI better equipped for unforeseen challenges in evolving environments, potentially spurring beneficial research in adaptive systems, it also introduces complexities. Potential risks include the misapplication of harnessing volatility without stringent safety protocols, the resource demands of developing such systems, and the uncertainty of how highly adaptive AI might evolve, particularly if not perfectly aligned with human values. Significant ethical considerations, such as responsible vulnerability disclosure, addressing potential biases in adaptive learning, and ensuring societal preparedness for dynamically evolving AI, must be proactively managed. Ultimately, this work seeks to stimulate critical discussion on dynamic, forward-looking strategies for AI safety, acknowledging the inherent uncertainties in developing truly antifragile systems while underscoring the importance of such a pursuit for trustworthy AI.

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

# A. Relation to Existing Dynamic Frameworks in AI

Many existing research areas grapple with changing or uncertain environments, and thus share some overlap with our call for an antifragile perspective. However, important conceptual differences remain, as we summarize below.

**Robust MDPs and Distributionally Robust Optimization.** These methods typically assume a known set of plausible transitions or reward perturbations (an uncertainty set) and aim to optimize worst-case performance within that set. An antifragile approach goes beyond simply minimizing worst-case loss within a fixed set of perturbations. Instead, it **invites unanticipated stressors**—states or transitions outside any pre-defined uncertainty set—and leverages them as opportunities to *expand* the model's domain of competence. Once a new stressor appears, antifragile systems grow from it (e.g., updating the threat model, incorporating new data, refining beliefs), rather than merely maintaining performance within a static boundary.

**Online Learning with Nonstationary Rewards (e.g., Bandits).** Classical online learning, including adversarial or nonstationary bandits, seeks to minimize regret in the face of changing reward distributions. While regret minimization indeed resonates with antifragile principles, most online-learning algorithms treat new adversarial patterns primarily as negatives to be mitigated. They do not typically *gain* from the shock, i.e., incorporate lessons that *widen* future safe performance boundaries or strengthen adaptivity across multiple dimensions. This aspect has been critically examined in the lower bound analysis in (Jin, 2024). In contrast, antifragile systems explicitly regard disruptions as vaccinations, using stress events to achieve net-positive adaptations for subsequent encounters.

**Meta-Learning, Out-of-Distribution Adaptation, & Continual Learning.** Meta-learning seeks to quickly adapt to new tasks by learning a good prior. Likewise, OOD adaptation focuses on bridging training–deployment gaps by adjusting to new data distributions, and continual-learning methods aim to sequentially update models without catastrophic forgetting. While crucial, these don't inherently involve actively seeking out and strengthening against extreme, safety-critical edge cases or black swan events the way an antifragile system would aspire to. They primarily aim to maintain performance or adapt smoothly, not necessarily to emerge stronger specifically from high-impact, rare failures by expanding the safety boundary itself.

**Adversarial Training, RLHF, and Iterative Refinement.** Techniques like adversarial training (Goodfellow et al., 2014; Madry et al., 2017) and Reinforcement Learning from Human Feedback (RLHF) (Ouyang et al., 2022) explicitly use failure cases (adversarial examples, undesirable outputs) to improve models. However, this is often done in discrete training cycles between model versions. This can lead to a reactive "whack-a-mole" dynamic where vulnerabilities are patched after discovery, rather than a continuous, proactive strengthening. Antifragility implies a more integrated, potentially real-time or near-real-time, mechanism where encounters with stressors directly trigger adaptation and generalization to related potential failures, aiming to reduce the *rate* at which new vulnerabilities appear.

**Internal Error Recovery via Backtracking.** One class of mechanisms enhancing resilience addresses *internal* failures detected during a system's ongoing process, such as logical errors in complex reasoning or safety violations during generation within LLMs. Approaches like Algorithm of Thoughts (AoT) and BSAFE implement structured *backtracking* capabilities (Sel et al., 2024; 2025a; Zhang et al., 2025; Sel et al., 2025b). The core idea is immediate error correction: when a flaw is detected mid-process, the system reverts to a previously known-good state and attempts an alternative execution path. This mechanism primarily aims to ensure the reliable or safe completion of the *current* task instance by recovering from specific, localized failures. Extensions using reinforcement learning to optimize this very recovery and exploration strategy (Sel et al., 2025c) demonstrate a potential link to antifragility, where the system learns to improve its problem-solving robustness by experiencing and overcoming internal errors.

**Self-Correction through Reflection on Outcomes.** Distinct from immediate path correction via backtracking, another family of techniques emphasizes *self-correction or reflection* based on evaluating *past* actions, completed outputs, or trajectory outcomes. Methods inspired by reflection, such as Reflexion agents (Shinn et al., 2023), allow a system to analyze its performance (e.g., task success/failure, quality of output) and use this evaluation ("reflection") as a learning signal. This feedback guides *future* attempts or refines the overall strategy for subsequent actions within the task context. From the perspective of this paper, reflection contributes significantly to learning and adaptation, but might primarily strengthen performance within existing boundaries, whereas antifragility also emphasizes the potential expansion of those boundaries when confronted with truly novel stressors or failures.

**Adaptive Filters and External Safeguards.** Some systems use dynamic external components, like evolving content classifiers (Sharma et al., 2025), to block harmful outputs. These act as adaptive shields but don't necessarily make the underlying core model itself antifragile; the model's internal understanding might remain static between

updates, and the safeguard only catches known or similar emerging attack patterns.

**Organizational Practices (e.g., Red Teaming).** Iterative red teaming (Ganguli et al., 2022) is a practical implementation of seeking out failures. However, its typically human-driven nature limits the speed and scale of adaptation compared to the ideal of an automated system capable of continuous self-improvement from encountered stressors.

*Summary of Distinction:* In essence, while many existing techniques contribute to robustness and adaptation, antifragility as proposed here involves a system-level commitment to (1) potentially actively (though safely) seeking stressors or treating unexpected events as primary learning signals, (2) using failures not just to patch but to systematically expand the safe operating regime and generalize against future novel threats, and (3) striving for continuous, potentially automated, adaptation loops rather than relying solely on discrete, often human-in-the-loop, update cycles. Our dynamic regret framework aims to capture this continuous process of reducing the gap between the system's perception and reality.

## B. Value and Probability Distortion Functions

**Definition B.1** (Reward Distortion Function (Lee et al., 2025)). The reward distortion function $u$ is defined as:

$$u(r) = \begin{cases} u^+(r) & \text{if } r \geq 0, \\ u^-(r) & \text{if } r < 0, \end{cases}$$

where $u^+ : \mathbb{R}_{\geq 0} \to \mathbb{R}_{\geq 0}$ is non-decreasing, concave with $\lim_{h \to 0^+}(u^+)'(h) \leq 1$, and $u^- : \mathbb{R}_{\leq 0} \to \mathbb{R}_{\leq 0}$ is non-decreasing, convex with $\lim_{h \to 0^-}(u^-)'(h) > 1$.

**Definition B.2** (Probability Distortion Function(Lee et al., 2025)). The probability distortion function $w$ is defined as:

$$w(p) = \begin{cases} w^+(p) & \text{if } r \geq 0, \\ w^-(p) & \text{if } r < 0, \end{cases}$$

where $w^+, w^- : [0,1] \to [0,1]$ satisfy: $w^+(0) = w^-(0) = 0$, $w^+(1) = w^-(1) = 1$; $w^+(a) = a$ and $w^-(b) = b$ for some $a, b \in (0,1)$; $(w^+)'(x)$ is decreasing on $[0,a)$ and increasing on $(a,1]$; $(w^-)'(x)$ is increasing on $[0,b)$ and decreasing on $(b,1]$.

## C. Proof of Corollary 3.4

*Proof.* Use the standard three-term decomposition:

$$\begin{aligned} &\left| V_{\mathcal{M}}(\pi^\dagger) - V_{\mathcal{M}}(\pi^\star) \right| \\ &= \Big| V_{\mathcal{M}}(\pi^\dagger) - V_{\mathcal{M}^\dagger}(\pi^\dagger) + \\ &\qquad V_{\mathcal{M}^\dagger}(\pi^\dagger) - V_{\mathcal{M}^\dagger}(\pi^\star) + V_{\mathcal{M}^\dagger}(\pi^\star) - V_{\mathcal{M}}(\pi^\star) \Big| \\ &\geq \left| V_{\mathcal{M}^\dagger}(\pi^\dagger) - V_{\mathcal{M}^\dagger}(\pi^\star) + V_{\mathcal{M}}(\pi^\dagger) - V_{\mathcal{M}^\dagger}(\pi^\dagger) \right| \\ &\qquad - \left| V_{\mathcal{M}^\dagger}(\pi^\star) - V_{\mathcal{M}}(\pi^\star) \right| \\ &\geq \left| V_{\mathcal{M}}(\pi^\dagger) - V_{\mathcal{M}^\dagger}(\pi^\dagger) \right| - \left| V_{\mathcal{M}^\dagger}(\pi^\dagger) - V_{\mathcal{M}^\dagger}(\pi^\star) \right| \\ &\qquad - \left| V_{\mathcal{M}^\dagger}(\pi^\star) - V_{\mathcal{M}}(\pi^\star) \right| \\ &= -\Big[ V_{\mathcal{M}^\dagger}(\pi^\dagger) - V_{\mathcal{M}^\dagger}(\pi^\star) \Big] + \Delta_1(\pi^\dagger) - \Delta_1(\pi^\star). \end{aligned}$$

The result follows by conditions (1)–(3). □

The result implies that the perceived-optimal policy $\pi^\dagger$ performs at least $c^\dagger - (\delta^\dagger + \epsilon^\star)$ points *worse* than $\pi^\star$ in the true environment. Thus, the suboptimality gap is strictly away from zero. This scenario naturally arises when $\pi^\dagger$ exploits illusions about high-reward or negligible-cost states that, in truth, occur with a small positive probability and incur catastrophic negative reward (black swans). As soon as $c^\dagger$ exceeds $\delta^\dagger + \epsilon^\star$, a strictly positive suboptimality gap emerges.

## D. Roadmap for Extension to Multi-Agent or Partially Observed Settings

While we have focused on a single-agent MDP with full observability, many real-world safety challenges involve multiple agents or partial observability (POMDPs). The same logic of inevitable blind spots and dynamic regret can extend as follows. For multi-agent systems, model each agent with its own policy and environment belief. Black swans can arise from emergent interactions; an antifragile approach would continuously update the joint or opponent models whenever new adversarial strategies appear. A POMDP can be viewed as an MDP over belief states. Unknown or mis-modeled observation probabilities still create catastrophic failures if those events are never updated. Hence, iterative expansions of the observation model—and safe exploration to reveal hidden states—mirror the same antifragile logic.

## E. Design Principles for Achieving Antifragility

Antifragility is more than iterative retraining or patching. Here we list a partial list of guiding features or mechanisms that move a system from mere resilience to genuine antifragility:

**Open-Ended Data Exploration:** Instead of relying solely on a fixed training set or known threat model, antifragile systems incorporate open-ended data streams—including adversarially constructed or rare-event examples—to continuously extend their representation of the environment. For instance, an AI-driven cybersecurity suite might automatically analyze near miss logs from novel intrusion attempts and introduce them into an expanded environment simulation, forcing future models to prepare for those new intrusion patterns.

**Adaptive Threat Modeling:** In robust control, one typically assumes a bounded set of disturbances. Antifragile design assumes new disruptions will appear outside existing bounds—and systematically updates the environment model (i.e., includes newly found vulnerabilities, black swan states, or emergent adversarial strategies). This contrasts with static certifications: once the system is shown robust for a known class of threats, it does not end testing but explicitly seeks out untested conditions.

**Proactive Stress Testing and Sandbox Mechanisms:** Antifragile architectures incorporate safe fail mechanisms or sandbox environments where novel stressors can be tested without catastrophic real-world consequences. Crucially, the system or the community controlling it does not shy away from introducing carefully contained shocks. These safe fail experiments are not a one-time exercise; they form a continuous regimen aimed at discovering new edges of the state space.

**Self-Monitoring and Alerts for Drift:** Traditional systems often degrade over time if the environment drifts away from training conditions. An antifragile system includes triggers that detect drifts or anomalies early, then actively engages in a policy update (e.g., re-optimizing or augmenting the model) to build new competencies. Unlike a basic resilience approach (which might simply revert to a stable fallback policy), antifragile systems incorporate the new drift data to reduce the chance of repeated surprises.

**Mechanisms for Learning from Partial or Rare Feedback:** Because black swan events can be extremely sparse, antifragile systems rely on creative data augmentation, imitation from near misses, or structured simulations (digital twins) to approximate learning signals. The hallmark is that each new surprise is systematically curated into the environment model or threat library, feeding iterative improvement.

Hence, while robust systems and standard iterative updates can maintain baseline performance under known perturbations, antifragile designs expand the horizon of safe operation by actively assimilating every discovered failure into an evolving threat or environment model.

# F. Concrete examples and empirical evidence

*Biological systems:* Evolution is inherently antifragile—pressures prompt adaptations enabling greater species resilience over generations, like bacteria developing stronger resistance to antibiotics designed to eliminate them. Tropism in plants allow dynamically bending towards beneficial stimuli like light, enhancing robustness despite variability. Even mild climate changes may elicit adaptive responses in coral resilience (Hughes et al., 2003). Ecosystems with higher biodiversity demonstrate greater adaptability to environmental changes (Folke et al., 2004), and can be actively leveraged to improve resilience, e.g., prescribed fire (Ryan et al., 2013). The immune system strengthens from exposure to pathogens (e.g., through vaccination (Plotkin, 2005)), and skeletal muscles grow in response to the moderate stress of exercise (Schoenfeld, 2010).

*Economic systems:* Decentralized markets, characterized by price fluctuations and competition, drive innovation and efficiency (Hayek, 2013). Entrepreneurship often benefits from failure and adversity, leading to future success (McGrath, 1999). Some investment strategies, such as antifragile portfolios like "long vega" and "long gamma" financial derivatives, are designed to profit from market volatility (Taleb & Douady, 2013).

*Social systems:* Collective intelligence, which relies on the diversity of opinions and experiences (viewed as internal opinion stress testing), enhances problem-solving and decision-making capabilities (Page, 2008). Resilient communities adapt and thrive under challenge (Norris et al., 2008).

*Technological systems:* Agile development allows rapid response to changing requirements (Manifesto, 2001).

*Psychology:* Adversity can lead to post-traumatic growth (Tedeschi & Calhoun, 2004). Hormesis (Mattson, 2008), where low doses of toxins trigger beneficial responses, is another example.

*Engineering systems.* Early steam engines advanced from fragile explosiveness to reliable operations due to an engineering discovery—intentionally introducing randomness (dithering) stabilized operations by overcoming mechanical stiction. Chaos Control theory explores how the principles of chaos theory can be applied to engineering systems to achieve faster control and stability (Ott et al., 1990). The concept of using noise for stabilization in early steam engine design, known as "stochastic resonance" or "noise-induced stability," is related to the principles of Chaos Control theory (Gammaitoni et al., 1998). By understanding and leveraging the properties of chaotic systems, engineers can design more efficient and responsive control systems that are agile, adaptive, and capable of rapidly responding to changes in their environment (Schöll & Schuster, 2008).

While some sources may use terms like "resilience," these examples go beyond mere recovery. Under stress, a resilient system bounces back; an antifragile system bounces forward, returning to a state stronger than before. This distinction highlights the potential benefits of designing with antifragility in mind.

## G. Feasibility, Cost, and Risk in Critical Applications

In high-stakes fields such as healthcare, finance, or critical infrastructure, deliberately introducing new live failures or stressors can be both risky and ethically fraught:

**Safe Sandboxes and Simulations:** The best practice is to use realistic digital twins, simulation platforms, or carefully isolated test wards (in healthcare) or test networks (in power grids) where catastrophic outcomes do not harm real stakeholders. While building and maintaining such simulators is resource-intensive, it is essential for antifragile testing and is increasingly common in industries like aerospace and autonomous driving.

**Phased Rollouts and Controlled A/B Testing:** When real-world testing is unavoidable, organizations can gradually deploy updates to a small user group or in non-critical use-cases first. This phased rollout approach balances the need for exposure to real conditions with risk mitigation. Monitoring near-misses or anomalies in these subsets can yield valuable data for new environment states without endangering the entire system at once.

**Resource Constraints and Smaller Labs:** Not every organization can afford large-scale, continuous requalification. Open-source tools and shared testbeds (akin to adversarial ML challenge platforms) can help democratize access to stress testing. Encouraging a collaborative ecosystem—where vulnerabilities or novel attacks are responsibly disclosed and integrated into publicly available test suites—helps less resource-rich players benefit from the community's collective knowledge.

Ultimately, while antifragility does involve cost and risk, it need not be done blindly or recklessly. Thoughtful sandboxing, staged testing, and well-designed simulations allow systems to gain from adversity without inflicting undue harm.

## H. Disclosure and Collaboration

**Timing and Scale of Vulnerability Disclosure** A critical aspect of antifragile AI practice is deciding when and how to share newly discovered vulnerabilities or failure modes. While fully transparent disclosure can accelerate collective learning, it also risks exposing exploitable weaknesses before fixes are in place. In practice, many fields (e.g., cybersecurity) use responsible disclosure processes: researchers privately inform affected parties about a flaw, provide a short window for remediation, then publicly announce the vulnerability—ideally alongside a recommended patch. This approach strikes a balance between safety (minimizing immediate exploitation) and community benefit (allowing the broader ecosystem to learn and adapt).

In especially sensitive domains (e.g., nuclear systems, critical infrastructure), coordinated release may be required—multiple agencies or organizations agree on embargo periods, partial data releases, and mutual assistance in remediation. Although slower, such coordination ensures patch readiness across different stakeholders before public announcements prompt malicious exploitation.

**Balancing Community Collaboration with Proprietary Constraints** Despite the collective advantages of sharing vulnerabilities, many safety-critical sectors operate under tight confidentiality (e.g., finance, defense, medical records). Full transparency may be impossible due to regulatory or commercial concerns. In such cases, trusted consortia can foster safe, limited-scope disclosure: relevant organizations, possibly under non-disclosure agreements, circulate sanitized attack signatures or emergent threat patterns without exposing proprietary data. This lets each participant incorporate newly revealed exploits into their antifragile pipeline—updating threat models, refining simulations, and bolstering overall resilience.

Even outside formal consortia, abstracted taxonomies of discovered failures can be made publicly available. For example, a bank might not disclose specific transaction logs but can publish high-level exploit categories (e.g., cross-service token forging or injection via unverified API bridging). Over time, such taxonomies enable both large and smaller labs to benefit from each other's stressors and reduce duplicated vulnerabilities. Designing incentives—like bug bounties, recognition, or regulatory credits—can further motivate organizations to collaborate on this shared goal of building antifragile AI systems.

