# OpenReview forum: "Position: AI Safety Must Embrace an Antifragile Perspective"
_ICML.cc/2025/Position_Paper_Track — ICML 2025 Position Paper Track poster_

### Official Review · Reviewer_yQPS · 2025-03-12

**Significance:** 3
**Argument Clarity:** 3
**Rating:** 4
**Confidence:** 3

**Questions:**

1. Could you provide a brief explanation of how MDPs specifically relate to antifragility in AI safety?
2. Why does the paper focus mainly on black swan events rather than more frequent but still novel failures that fall outside the system’s current model?
3. Could you provide a more direct comparison with current adaptive safety mechanisms and discuss why they are insufficient for achieving antifragility?

**Discussion Potential:**

3

**Paper Summary:**

This paper argues that AI safety research should adopt an antifragile perspective, emphasizing systems that improve their capacity to handle rare or out-of-distribution (OOD) events through repeated exposure. The authors critique conventional AI safety methods that rely on static benchmarks and single-shot robustness tests, which may lead to maladaptation over time, such as reward hacking or over-optimization. They advocate for dynamic safety approaches that leverage uncertainty rather than merely reducing it. The paper identifies key challenges, including scenario diversity, over-alignment, and reward hacking, and proposes guidelines for measuring and improving AI safety in an evolving environment.

## Update after rebuttal

The paper presents an engaging and timely position on antifragility in AI safety, challenging the field’s prevailing emphasis on static robustness. Its critique of static benchmarks and the call for continuous, self-improving systems is well-articulated and likely to stimulate valuable discussion in the community. In response to my concerns, the authors have provided a thoughtful clarification on their use of MDPs, framing them as a high-level abstraction for modeling the gap between perceived and real environments. While this helps contextualize their approach, a brief introduction in the paper itself would still benefit readers unfamiliar with reinforcement learning. Their justification for focusing on black swan events is more persuasive after the rebuttal, though the addition of explicit criteria for antifragile challenges could still improve clarity. Importantly, the authors responded in detail to the critique about related work, offering a strong comparison with adaptive methods like meta-learning, continual learning, and adversarial training. This significantly strengthens the position that antifragility is distinct from incremental robustness. Given the authors’ clear responses and plans to incorporate these points, I believe the paper makes a compelling contribution to the conceptual landscape of AI safety. I increased my score.

**Position:**

Yes

**Position In Title:**

Yes

**Related Work:**

2

**Strengths And Weaknesses:**

Strengths:

* Timely and Relevant Topic: The discussion on antifragility in AI safety is novel and relevant, especially as AI systems are increasingly deployed in open-ended, evolving environments.

* Critique of Static Benchmarks: The paper effectively highlights the limitations of conventional robustness evaluations, which often fail to account for long-term adaptation and drift.

* Potential for Discussion: The position is likely to spark meaningful debate in the AI safety community, encouraging alternative perspectives on safety evaluation.

Weaknesses:

* Lack of Background on MDP:

The paper frequently references Markov Decision Processes (MDPs) but does not provide sufficient background. A brief introduction to how MDPs relate to antifragility and AI safety would improve clarity, particularly for readers unfamiliar with reinforcement learning.

* Overemphasis on Black Swan Events:

The discussion primarily links antifragility to black swan events—rare, high-impact occurrences. However, other more frequent but still unforeseen events (e.g., gradual shifts in data distributions) can also cause system failure.

It would strengthen the argument to explicitly justify why black swan events are the primary focus. One possible approach is to define a lower bound for events that qualify as antifragile challenges.

* Insufficient Discussion of Existing Work:

While the paper critiques static robustness tests, it does not provide an in-depth comparison with existing adaptive safety methods (e.g., continual learning, robustness against distribution shift).

A more structured discussion on why existing AI safety approaches do not adequately support antifragility would improve the argument.

**Support:**

3

---

> ### Author Rebuttal · Authors · 2025-03-31
>
> We thank the reviewer for the valuable feedback. Below, we clarify our responses in three parts.
>
> **1. MDPs and Antifragility**
>
> Our paper uses MDPs as a high-level abstraction to formalize a central challenge in AI safety: the gap between a system’s internal (perceived) model and the real environment. By modeling the sequential, interactive nature of AI systems—similar to approaches in performative prediction—MDPs allow us to quantify this “robustness gap” via dynamic regret. In our formulation, antifragility emerges from a two-timescale process: a fast loop that optimizes the policy based on the current (and possibly flawed) model, and a slow loop that updates the model to better align with reality. Even for readers not deeply familiar with reinforcement learning, the takeaway is that MDPs allow us to mathematically capture how static, one-time safety measures leave systems vulnerable to unforeseen failures.
>
> **2. Black Swan Focus**
>
> We emphasize black swan events not to exclude more frequent novel failures but to highlight a worst-case scenario that exposes the limitations of static defenses. Black swan events represent those rare, high-impact occurrences that lie outside a system’s current threat model. In practice, the iterative feedback from more frequent failures serves to progressively narrow the robustness gap, enhancing the system’s capacity to handle even catastrophic events. Thus, our focus on black swan events underscores the critical need for continuous, proactive adaptation as a complement to addressing more moderate, predictable failures.
>
> **3. Comparison with Current Adaptive Safety Mechanisms**
>
> While current adaptive safety methods contribute to AI robustness, they fall short of achieving true antifragility for several reasons.
> First, approaches in meta-learning (e.g., MAML [1]) and continual learning [2] enable adaptation to new tasks and help prevent catastrophic forgetting, but they generally incorporate new data passively rather than actively seeking out extreme edge cases that challenge the system’s safety limits [3]. Second, adversarial training and RLHF update models using failure cases (such as jailbreak examples) between model versions—resulting in a “whack-a-mole” effect where each vulnerability is patched only after it appears, rather than continuously strengthening the model in real time. Third, methods like Reflexion agents [4] leverage self-feedback to improve performance within an episode, yet they remain confined to predetermined task boundaries and do not explore beyond their current operating regime. Fourth, adaptive filters, such as Anthropic’s evolving classifiers [5], dynamically update as new attack vectors emerge; however, these function as external safeguards while the core model remains static between updates. Finally, organizational practices like red-teaming and iterative model releases are valuable for identifying vulnerabilities, but the inherent human-driven cycles result in slow adaptation that cannot match the demands of real-time safety in open-ended environments.
>
> True antifragility requires that systems do not merely resist failures, but actively learn from them. This involves:
> * Actively seeking out stressors: Systems must deliberately probe their vulnerabilities rather than passively absorbing new data.
> * Using failures to expand safety: Instead of merely maintaining existing safety boundaries, each failure should trigger an expansion of the safe operating regime—a process our dynamic regret formulation quantifies.
> * Continuous adaptation: Rather than relying on discrete cycles of training, deployment, and retraining, antifragility demands real-time learning that strengthens the system with every exposure to risk.
>
> In short, while existing methods incrementally improve robustness, they do not support the full vision of antifragility, which requires a system-level approach to proactive, scalable self-improvement.
>
> References:
> [1] Finn, C., Abbeel, P., & Levine, S. (2017, July). Model-agnostic meta-learning for fast adaptation of deep networks. In International conference on machine learning (pp. 1126-1135). PMLR.
> [2] Parisi, German I., et al. "Continual lifelong learning with neural networks: A review." Neural networks 113 (2019): 54-71.
> [3] Khattar, V., Ding, Y., Sel, B., Lavaei, J. and Jin, M., A CMDP-within-online framework for Meta-Safe Reinforcement Learning. ICLR 2023
> [4] Shinn, N., Cassano, F., Gopinath, A., Narasimhan, K. and Yao, S., 2023. Reflexion: Language agents with verbal reinforcement learning. Advances in Neural Information Processing Systems, 36, pp.8634-8652.
> [5] Sharma, Mrinank, et al. "Constitutional classifiers: Defending against universal jailbreaks across thousands of hours of red teaming." arXiv preprint arXiv:2501.18837 (2025).

---

### Official Review · Reviewer_gE2g · 2025-03-14

**Significance:** 3
**Argument Clarity:** 3
**Rating:** 3
**Confidence:** 3

**Questions:**

While the theoretical arguments are strong, the paper lacks real-world experiments or empirical studies showing antifragile AI in action.
Including case studies on antifragile AI systems against existing robustness methods would strengthen the argument.

**Discussion Potential:**

3

**Paper Summary:**

This position paper argues that AI safety should adopt an antifragile approach—meaning AI systems should not only focus on safety and robustness studies but improve from exposure to black swans. The authors critique the current robustness paradigm, which relies on static benchmarks and one-time safety checks, making AI vulnerable to evolving threats such as reward hacking, over-alignment, and adversarial attacks. Instead, they advocate for a time-evolving, self-improving safety framework that treats uncertainty as a learning opportunity rather than a mere risk to be mitigated.

**Position:**

Yes

**Position In Title:**

Yes

**Related Work:**

3

**Strengths And Weaknesses:**

The paper challenges mainstream AI safety approaches, proposing a more adaptive and evolving framework.
The authors provide formal definitions of black swan events and robustness gaps, grounding their arguments in mathematical models.
The discussion on LLM jailbreaks, adversarial attacks, and cybersecurity risks makes the paper highly relevant to modern AI safety challenges.

**Support:**

3

---

> ### Author Rebuttal · Authors · 2025-03-31
>
> Thank you for highlighting the importance of empirical validation. While our paper is primarily theoretical in nature, we agree that empirical evidence strengthens our position and offers important context for our framework.
>
> For instance, the Adversarial NLI (ANLI) benchmark demonstrates that models trained over multiple adversarial rounds significantly outperform standard models when facing novel challenges [1]. Similarly, platforms like Dynabench—by dynamically collecting human-generated adversarial examples—illustrate a slow-loop adaptation process that yields incremental safety gains [2]. Moreover, industry practices at organizations such as OpenAI and Anthropic, which employ extensive red-teaming and targeted training, provide early evidence that systems can benefit from iterative, reflective updates.
>
> Our framework builds on these observations by advocating for a continuous, automated reflective loop that updates the model’s internal threat representation. This approach shifts the focus from merely resisting known disruptions to actively learning from each encountered stressor, thereby reducing the gap between perceived and actual risks.
>
> We acknowledge that antifragility in AI safety remains an emerging area that calls for further empirical investigation. One key research direction is to develop methods that not only automate the reflective loop but also effectively mitigate distributional shifts. Drawing upon the experience of Dynabench, further work is needed to design ensemble-based approaches, incorporate diverse human validations, and establish data retention strategies that maintain a balanced threat model even as new adversarial data are incorporated. Another promising direction involves formulating and empirically validating metrics such as dynamic regret reduction, which can quantify how well a system improves over successive rounds of exposure to novel challenges. Additionally, exploring hybrid models that seamlessly integrate human-in-the-loop feedback with automated adaptation could further enhance scalability and robustness.
>
> In summary, while existing empirical work and industry practices provide valuable building blocks, our framework reorients the safety objective from merely resisting disruptions to growing stronger through them. Our theoretical framework and practical guidelines (Section 4.2) offer a roadmap for developing system-level, antifragile safety measures.
>
>
> References:
> [1] Nie, Y., et al. "Adversarial NLI: A New Benchmark for Natural Language Understanding." Proceedings of the 58th Annual Meeting of the Association for Computational Linguistics. 2020.
> [2] Meta AI's Dynabench: https://dynabench.org/

---

> > ### Comment · Reviewer_gE2g · 2025-04-04
> >
> > Thanks for the clarification. After reading the rebuttal, I would like to keep my score.

---

### Official Review · Reviewer_drJo · 2025-03-14

**Significance:** 4
**Argument Clarity:** 3
**Rating:** 4
**Confidence:** 3

**Questions:**

Please respond to the weaknesses above.

**Discussion Potential:**

3

**Paper Summary:**

The authors argue that AI safety research should be viewed from a time evolving perspective. Instead of having static robustness tests and benchmarks, the authors argue for time-evolving benchmarks and performance evaluations, as well as the need for antifragile systems, which dynamically adapt to distribution shifts to become more resilient. The authors formulate the problem as an MDP, and show that in multi-step environments, there exist black swan events where the agent's policy is highly suboptimal. Then, the authors define anti-fragile systems formally using a regret-based definition.

## update after rebuttal
I have gone through the rebuttal and the comments from other reviewers. Most of my concerns have been addressed and I would like to keep my score.

**Position:**

Yes

**Position In Title:**

Yes

**Related Work:**

2

**Strengths And Weaknesses:**

Strengths:
- The message of the paper is very timely, and connects to real-world examples (e.g. jailbreaking)
- The concept of anti-fragile systems (and contrasting with robust or resilient systems) is under-explored in the field.
- The authors use an MDP-based formulation to more rigorously define black swan events and antifragility.
- The authors give concrete red flags, and concrete next steps towards antifragility.

Weaknesses:
1. Most of the mathematical formulation and theorems are reproduced or easily adapted from Lee et al. (2024) and Jin (2024), and so the novelty in the formulation is limited.

2. The argument that "black swan events are inevitable" (Section 2.2) seems very hand-wavy relative to the rest of the formulation. Is it possible to come up with a theorem which lower bounds the probability of such an event?

3. The discussion of current black swans in LLMs (Section 2.3) is focused solely on jailbreaking or adversarial prompts. Are there other concrete examples which have lead to greater real-world impact?

4. The authors should discuss if/how current research in LLM adaptation/robustness may be useful for antifragility, e.g. test-time adaptation, model editing.

**Support:**

3

---

> ### Author Rebuttal · Authors · 2025-03-31
>
> We appreciate your thoughtful feedback. Below we address your specific concerns:
>
> **1. Mathematical Novelty and Adaptation from Prior Work.**
>
> While our formulation builds on results from Lee et al. (2024) and Jin (2024), our key contribution is reframing these results to expose a critical shortcoming of static defenses: the "robustness gap" that emerges when a system's internal model deviates from reality. This shift—from one-time certification to continuous safety improvement—is not captured by previous work. Our dynamic regret formulation quantifies how static defenses accumulate risk, motivating continuous model expansion.
>
> **2. Lower Bounding the Probability of Black Swan Events**
>
> To address your concern about the inevitability of black swan events, we offer the following formulation (adapted from Theorem 5 in Lee et al., 2024):
>
> Assume $P(s'|s,a) > 0$ for any $s, s' \in \mathcal{S}$ and $a \in \mathcal{A}$ under policy $\pi^\dagger_t$ at time $t$. Let a black swan event be defined as in Definition 2.1. Then, if the agent takes $t$ steps such that $t \geq \frac{\log(\delta/p_{\min})}{\log(1-p_{\max})} + 1$, where $p_{\min} = \frac{R_{\max}-R_{bs}}{2R_{\max}}\epsilon_{bs}^{\min}$ and $p_{\max} = \frac{R_{\max}-R_{bs}}{2R_{\max}}\epsilon_{bs}$, with $\epsilon_{bs}^{\min}$ being the minimum probability of encountering a state-action pair contributing to the robustness gap, $\epsilon_{bs}$ the threshold for "rare" probabilities in function $w$, $R_{bs}$ the threshold for severe reward misperception in function $u$, and $R_{\max}$ the maximum reward magnitude, the agent will encounter a black swan event with at least probability $\delta$.
>
> In simpler terms, this shows that even in stationary environments, catastrophic failures become increasingly likely unless the system updates its risk understanding.
>
> **3. Broader Examples of Black Swans**
>
> In our paper, we use jailbreaking as one illustrative example to highlight a fundamental issue: AI black swan events expose misalignments between anticipated and encountered risks. Additional examples include: (1) For instance, in 2023, ChatGPT generated fictitious legal citations that led to court sanctions [1]. This failure mode was not caught during pre-deployment testing, prompting subsequent adaptations such as improved warning systems and verification features. (2) The 2018 Uber self-driving car fatality demonstrating how perception systems fail with unforeseen edge cases [2]. (3) In cybersecurity, incidents such as the 2017 Triton malware attack and a 2024 CISA red team assessment—where a single overlooked web shell enabled complete domain compromise—demonstrate how misperceptions of risk can lead to catastrophic failures [3,4]. (4) Moreover, RL systems "hacking" their reward functions through unanticipated optimizations.
>
> Each example demonstrates our theoretical concept of a robustness gap—where the system's perceived model failed to account for critical real-world states.
>
> **4. Leveraging Current Adaptation/Robustness Techniques for Antifragility**
>
> Current research in LLM adaptation provides valuable building blocks for antifragile systems, but realizing true antifragility requires a shift from reactive to proactive strategies. For example, test-time adaptation (TTA) offers promising pathways toward antifragility by enabling real-time adjustments during deployment [5]. When faced with distribution shifts such as new topics or user behaviors, TTA-equipped LLMs can potentially transform performance degradation into opportunities for enhancement. Similarly, model editing techniques allow for precise knowledge updates without full retraining, while machine unlearning enables the strategic removal of outdated or biased information—each converting specific errors into discrete opportunities for improvement.
>
> An antifragile system integrates these techniques into a continuous adaptation loop that actively identifies weaknesses and updates the model's internal threat representation—shifting from post-hoc patches to a process that continuously expands safety boundaries.
> Challenges include developing adaptation mechanisms that preserve safety guardrails, creating efficient verification systems, and designing frameworks for active exploration of limitations. Our framework reorients safety from merely resisting disruptions to growing stronger through them.
>
> References:
>  [1] Mata v. Avianca, No. 22-cv-1461 (S.D.N.Y. June 22, 2023).
>  [2] "How a Self-Driving Uber Killed a Pedestrian in Arizona" (New York Times, March 2018).
>  [3] "Enhancing Cyber Resilience: Insights from CISA Red Team Assessment of a US Critical Infrastructure Sector Organization" (CISA, November 2024).
>  [4] "TRITON Malware Remains Threat to Global Critical Infrastructure Industrial Control Systems" (IC3, March 2022).
> [5] Xiao, Z. and Snoek, C.G., 2024. Beyond model adaptation at test time: A survey. arXiv preprint arXiv:2411.03687.

---

### Official Review · Reviewer_XYsu · 2025-03-17

**Significance:** 3
**Argument Clarity:** 3
**Rating:** 3
**Confidence:** 4

**Questions:**

+ The authors say that they adapted Theorems 2.2 and 2.3 from (Lee et al., 2024c), but they did not provide a formal proof of these theorems. What is the difference between Theorems 2.2-2.3 and the work in (Lee et al., 2024c)? How do the proofs of these theorems differ from (Lee et al., 2024c)?

**Discussion Potential:**

3

**Paper Summary:**

This position paper advocates that AI safety research should adopt an antifragile approach, where AI systems improve when exposed to rare and unexpected events rather than simply resisting them. The paper provides a theoretical foundation, through the lens of Markov Decision Processes (MDPs), showing that, under this perspective, black swans are inevitable, and outlines some practical and ethical guidelines for an antifragile approach to AI safety.

**Position:**

Yes

**Position In Title:**

Yes

**Related Work:**

3

**Strengths And Weaknesses:**

Strengths:
+ The paper provides an interesting approach to AI safety through the lens of Markov Decision Processes, with an original argument that AI safety should move beyond robustness and embrace an antifragile approach, where the system learns and improved after being exposed to rare events. I believe this argument can open the discussion on how to tackle AI safety from a wider perspective than traditional robustness-based approaches.
+ The paper includes a nice theoretical work showing how black swan events are inevitable for AI system, which can lead to catastrophic failures under current AI safety paradigms.
+ The authors discuss and explore possible avenues for implementing an antifragile approach to AI safety, like dynamic stress testing, adversarial feedback loops, or periodic policy reviews.


Weaknesses:
+ The paper defends antifragility as the posture to be taken for dealing with AI safety instead of other views like traditional robustness or resilience. However, AI systems are very diverse and, in some contexts, a robustness or a resilience-based approach can be more suitable. For example, in applications with low concept drift where the models are quite static over time robustness can provide useful insights on models’ safety. In other applications, like robotics for industrial environments, resilience is also adequate. In contrast, for LLMs and generative AI models, an antifragile would be quite beneficial. In this sense, I think the paper lacks a more thorough discussion about in which contexts each one of these approaches is more reasonable. On the other hand, I do not think that the assumption of treating an AI system as an MDP is practical in all scenarios.
+ The paper provides compelling theoretical arguments but lacks a more practical perspective to show how antifragile AI methods can be implemented effectively. In this sense, the proposed antifragility measures remain abstract.
+ Following the previous point, the paper does not discuss sufficiently the practical challenges and limitations of deploying the proposed strategies in practical scenarios.
+ I think that the paper presents a skeptical position on robustness techniques traditionally used in AI safety, which provide valuable risk assessment and mitigation in many domains. However, many of the proposed solutions to take this antifragile approach, really rely on robustness-based techniques. The difference, perhaps, is the continuous monitoring an adaptation of systems to make them improve over time.


Minor comment:
+ MDP (Markov Decision Process) acronym is never introduced in the paper.

**Support:**

3

---

> ### Author Rebuttal · Authors · 2025-03-31
>
> We thank the reviewer for their thoughtful feedback. We address each key point below:
>
> **1. Use of Theorems from (Lee et al., 2024c)**
>
> We leverage results (from Lee et al., 2024c) to formalize the concept of a “robustness gap” that underlies black-swan failures. While the core ideas remain consistent with Lee et al., our theorems extend their work by reframing the gap in terms of dynamic regret—quantifying how static, one-time safety measures inevitably leave systems vulnerable unless the threat model is continuously updated. This naturally leads to a two-timescale process: a fast loop that optimizes the policy within the current model, and a slow loop updates the model to better align with real-world conditions. Our paper does not provide full proofs of these theorems since they follow the established structure of Lee et al.; our focus is on reframing these results to advocate for a paradigm shift. Unlike the static analyses in Lee et al., our formalism provides a path toward measuring continuous safety improvement and demonstrates that every unmodeled stressor accumulates risk unless the system actively learns from it.
>
> **2. Context-Dependent Safety Approaches**
>
> We agree that different applications require different safety approaches. Our position is not that antifragility should replace robustness or resilience in all contexts, but rather:
> * For stable, closed environments (industrial robotics, low-concept-drift applications): Traditional robustness methods are appropriate and sufficient
> * For open-ended, evolving domains (LLMs, cybersecurity): Static defenses leave systems vulnerable to unknown threats, making antifragility essential
>
> Our framework offers a complementary perspective that becomes increasingly important as AI systems operate in more complex, dynamic environments.
>
> **3. Practicality of MDP Modeling**
>
> We recognize the reviewer's concern about modeling all AI systems as MDPs. We use MDPs not as a literal implementation model but as an analytical tool that captures a universal challenge in AI safety: the mismatch between a system's internal model and the actual environment. This abstraction helps illustrate why continuous updating is necessary, regardless of the specific AI architecture.
>
> **4. Practical Implementation**
>
> The proposed metrics (dynamic regret and robustness-gap shrinkage) aims to measure long-term safety improvements. Our goal is to establish clear principles that serve as a roadmap rather than a complete technical solution. We recognize real-world constraints—particularly safety and resource limitations. Our guidance in Section 4.2 (e.g. checklists, red flags, structured updates) aims to concretize how one can embed continuous monitoring and iterative stress testing into the ML system, enabling it to adapt and generalize proactively. The goal is to gradually offload or assist/augment the efforts of some of the long-timescale safety analysis from humans to the system itself.
>
> **5. Relationship to Robustness Techniques**
>
> We acknowledge that antifragile strategies build on robustness methods. The key distinction is that robustness defends against known threats within fixed boundaries, while antifragility transforms each new stressor into an opportunity for expanding the system's safe operating regime. In other words, robustness is about whether a single model is robust to a given perturbation, while antifragility is about whether the system/adaptation process can adapt and grow when confronted with evolving threats.
>
> Our framework advocates for a shift from "patch-and-pray" cycles to continuous, proactive adaptation. This leads to several open problems, including: formally quantifying and minimizing the dynamic regret between a system's internal threat model and real-world conditions; developing mechanisms that trigger structured exploration of adjacent vulnerabilities upon encountering rare failures rather than just reactive patching; and designing protocols to safely expose deployed systems to novel adversarial challenges so that each "shock" contributes to future safety improvements.
>
> In sum, antifragility as a system goal requires robust foundations plus mechanisms to integrate and generalize from new threats—a vision that existing techniques, when used alone, do not fully support.

---

### Decision · Program_Chairs · 2025-05-01

**Decision:**

Accept (poster)

**Comment:**

Four expert reviewed the paper, and all rated the paper above the acceptance threshold. The reviewers liked the analysis through the lens of Markov decision processes in general except that one reviewer was less excited because this approach is largely borrowed from existing works. The reviewers acknowledged that the paper's position is clear and informative, shedding light on future work. Regarding the weaknesses, some reviewers were worried about the paper's disconnection from practice, lack of discussion of existing methods and the proposed position, and lack of insights for future directions to address the proposed position. The rebuttal is solid for answering these questions from the AC's view.